# Impact of Belgian COVID-19 lockdown restrictions on autistic individuals' socio-communicative behaviors and their parents' quality of life

Marielle Weyland[1,2]*, Pauline Maes[2], Mikhail Kissine[2], Pierre Defresne[3]

**1** Service Métrologie et Sciences du Langage, Université de Mons, Mons, Belgium, **2** ACTE at LaDisco and ULB Neuroscience Institute, Université libre de Bruxelles, Brussels, Belgium, **3** Fondation SUSA, Université de Mons, Mons, Belgium

* marielle.weyland@ulb.be

## Abstract

### Background

In the spring of 2020, Belgian authorities enforced a full lockdown period to contain the spread of the SARS-CoV-2 virus. This lockdown drastically disrupted the daily life of autistic individuals' and that of their families. In the midst of these extraordinary circumstances, we assessed the impact of social restrictions on autistic individuals' behavior and their parents' or caregivers' quality of life; we also sought to identify individual characteristics that may influence such changes.

### Methods

We designed an online survey targeting caregivers living with an autistic child or adult. The questionnaire included 125 five-point Likert questions which targeted changes in families' quality of life and in autistic individuals' behavior, as well as factors likely to influence the extent and direction of these changes.

### Results

We collected data from 209 French-speaking Belgian respondents. Respondents reported that the lockdown brought about a higher frequency of nonfunctional socio-communicative behaviors, as well as a decrease in families' quality of life. Parents who had less access to respite care experienced a steeper decrease in their quality of life. Autistic individuals with comorbidities, and whose parents had less access to respite care and implemented fewer rules at home during lockdown were more likely to display nonfunctional socio-communicative behaviors.

### Conclusion

COVID-19 lockdown restrictions had a negative impact on both autistic individuals and their parents.

**Data Availability Statement:** The minimal de-identified underlying data set needed to replicate all findings reported in this paper is available on the

Open Science Framework (https://osf.io/nzcbj/?view_only=ed6c70fc7abf4fee9d1be7ea376cc307), along with the questionnaire addressed to participants.

**Funding:** This work is supported by various sources of funding. MW and PD are supported by research grants from the Support Fund Marguerite-Marie Delacroix. PM is supported by a doctoral grant from the ROGER DE SPOELBERCH Foundation. MK is a 2019-2022 Francqui Foundation Research Professor. The funders had no role in study design, data collection and analysis, decision to publish, or preparation of the manuscript.

**Competing interests:** he authors have declared that no competing interests exist.

# Introduction

From spring 2020 onwards, many states worldwide enforced full lockdown periods to contain the spread of the SARS-CoV-2 virus. In Belgium, all residents, with the exception of health professionals and those working in services of prime necessity, were required to stay home between the 18th of March and 10th of May 2020. Belgium's National Security Council did allow outside physical activity, within a limited distance from one's residence, and for groups of maximum two people respecting physical distancing. On April 6th, these rules were partially relaxed for caregivers of individuals with a physical or mental disability, who were allowed to use their car for short non-essential trips, such as a walk in a park. Schools and kindergartens remained closed throughout the full lockdown period, but childcare services were made available for children of parents employed in prime necessity services and who could not work from home. Lockdown thus compelled most families to balance full-time parenting, which included homeschooling for some, with work arrangements and additional strains entailed by the lack of respite care and movement restrictions [1].

Outside the pandemic context, raising and caring for a child with a diagnosis of Autism Spectrum Disorder (ASD) leads to additional anxiety, as compared to families without an autistic child [2] or to families with a child with another disability [3]. Parents of autistic children report lower Quality of Life (QoL), reduced well-being and higher levels of stress [4–7], independently of the age of their child [8]. The World Health Organization defines the QoL as a multidimensional broad concept, which includes perception of one's physical and psychosocial state, and interpersonal relationships and social roles [9]. In addition, satisfaction with one's parental life is influenced by individual and environmental factors, which can obviously be impacted by such stressful events as the COVID-19 pandemic and lockdown restrictions.

Recent studies indicate that pandemic-related upheavals severely impacted the QoL and well-being of parents of autistic children in Northern Italy [10], in Turkey [11] and in Michigan [12]; these results also emerged from a comparative study [13] and a scoping review [14]. Surveys [15–17], parents and clinicians' testimonies [18, 19], and self-reports [20, 21] further attest to the impact that lockdown had on autistic individuals.

On the one hand, lockdown is likely to negatively impact autistic individuals who are highly sensitive to changes and attached to routines and environmental predictability [22–24], as well as those among them who need outdoor activities [25] or frequent intervention support. On the other hand, staying at home may suit those who seek self-isolation [26], relieving them from the usual social demands [21, 27]. Most pandemic-related effects previously reported in the autism literature go in the negative direction, but a decrease of psychopathological symptoms has also been reported [28, 29]. A recent systematic review by Alonso-Esteban and colleagues [30] also found that the consequences of social confinement may go both ways, leading them to advocate for more research on the short- and long-term psychological effects of lockdowns on individuals with special needs and their families.

In addition to describing the consequences of the lockdown, it is also important to identify which factors may influence these consequences. Several sets of factors are well-known to impact both autistic individuals' behaviors and their parents' QoL: these include family functioning, social and professional support [31, 32], severity of autistic behaviors [33], comorbidities [34] and educational and coping strategies [35]. During lockdown, modifications in autism symptomatology may also play a role on parents' QoL [36].

## Present study

The online survey reported below extends this line of research by investigating the impact the spring 2020 lockdown had on French-speaking Belgian autistic individuals and their families.

In line with studies that addressed the impact of COVID-19 in other countries, our study targets the QoL of families confined with their autistic child [11] and potential changes in the behaviors of the autistic individuals [10, 16]. In addition, however, we also investigate the potential external factors that may influence the extent and the direction of changes induced by the lockdown. As explained in the previous paragraph, these factors are based on individual characteristics that are particularly likely to impact both the autistic individuals' behaviors and their parents' QoL: family functioning, social and professional support, severity of autistic behaviors, comorbidities and educational and coping strategies.

In sum, our study sought to provide new insights into the short-term effects that lockdown periods had on autistic individuals and their families. We had three objectives. First, we sought to investigate the impact of lockdown restrictions on the autistic individuals' nonfunctional socio-communicative behaviors (i.e., behaviors that aim to obtain attention and/or to express positive or negative feelings without using the adequate mode of verbal or non-verbal communication) and on their parents' QoL. In doing so, we wanted to extend the findings reported for other countries [30] to French-speaking Belgian residents. In line with the literature, we expected that lockdown would have a negative impact on parents' QoL. We also hypothesized that lockdown would have an impact on the autistic individuals, but that this impact could be positive for some of them (e.g., those who seek self-isolation) and negative for others (e.g., those highly attached to routines). Second, we also wished to explore the relationship between the impact of lockdown on autistic individuals' nonfunctional socio-communicative behaviors and their parents' QoL. We expected a stronger decrease in parents' QoL when their autistic child experienced an increase in nonfunctional socio-communicative behaviors due to the lockdown restrictions. Finally, we also investigated the factors that may predict an improvement or deterioration of these behaviors and the parents' QoL in four domains: (a) household characteristics; (b) child characteristics; (c) financial and support resources and (d) home educational strategies. Our prediction was that the impact of lockdown would be more negative for families living in smaller homes, with younger or more severely disabled children or with less financial and support resources and lower access to educational strategies.

## Methods

The content and the procedure for this study were approved (16/04/2020) by the Faculty Ethics Advisory Committee of the Faculty of Psychology, Université Libre de Bruxelles, Belgium (Project ID: 094/2020). Prior to completing the survey, all participants provided electronic informed consent after being informed of the research aims and procedure, their rights and the contact information of the lead investigators.

We wanted to get the immediate reaction from our respondents on the effect of the lockdown, and for this reason, we decided to collect all information before the end of the lockdown. As a result, it was inevitable to restrict ourselves to families of autistic children whose members could take the time to answer an online questionnaire. That being said, our call for participation was very widely circulated across the French-speaking Belgian autistic community.

### Online questionnaire

A parent questionnaire (implemented in LimeSurvey, 3.22.7 [37]), was made available online between the 20[th] of April 2020, i.e. one month after the onset of the Belgian lockdown, and the 10[th] of May 2020, when the full lockdown ended. It was addressed to any primary caregiver living with an autistic child, adolescent, or adult. Inclusion criteria were being a Belgian French-speaking resident and primary caregiver living with an autistic child, adolescent, or adult.

Therefore, answers were retrieved from parents (and one grandparent) of autistic individuals, and not from autistic individuals themselves. Respondents were recruited via email lists from the ACTE lab's and the SUSA Foundation's internal databases, social media, and advocacy groups.

Prior to completing the survey, all participants provided electronic informed consent after being informed of the research aims and procedure, their rights, and the contact information of the lead investigators.

The survey was designed to collect in-depth information about each respondent's daily life during lockdown and their child's behavior during lockdown, as compared to their usual life. The core of the study is based on the parental evaluations of changes in their child's behavior and of their quality of life before and after lockdown. Questions were formulated to highlight these changes. That is, all questions targeting a change in behavior due to lockdown started with "Since lockdown,. . .?".

The full questionnaire included 125 questions: 19 yes/no questions, 60 5-point Likert Items (Much less than usual/A little less than usual/As usual/A little more than usual/Much more than usual), 45 unique or multiple-choice questions and 1 open-response question. The questionnaire probed 3 fields (family characteristics, your autistic child, your QoL during lockdown) divided into 15 subdomains (home organization, job, resources, daily life at home, your child's characteristics, your child's sleep, your child's diet, your child's socio-communicative behaviors, your child's progress, your emotional well-being, your material well-being, your physical well-being, your interpersonal relationships, your social connections, your personal development). The questionnaire took approximately 40 minutes to complete (which participants were made aware of before starting). Participants were allowed to stop at any point in the questionnaire and to return to it later. The questionnaire was structured to facilitate completion and to make it easy for participants to take breaks. The whole questionnaire is available on the Open Science Framework (https://osf.io/nzcbj/?view_only= ed6c70fc7abf4fee9d1be7ea376cc307).

Questions about QoL in our questionnaire were borrowed from the French validated version of the Parental-Developmental Disorders Quality of Life (Par-DD-QoL) [38]. Par-DD-QoL is a self-assessment questionnaire used to assess the impact of developmental disabilities on parental QoL in the following dimensions: emotional, daily disturbance and global QoL. However, we had to make some slight modifications to the questions in order to target changes due to the lockdown restrictions. First, each question was introduced by "Since the lockdown,. . ." or a similar formulation to target the changes in quality of life that were due to the lockdown. Second, because of this slight change in question formulation, we also had to adapt the answer modality so that it would fit a question targeting a change or an evolution in QoL and starting by "since the lockdown". For example, the following question and answer modality from the Par-DD-QoL:

Actuellement, du fait des troubles de votre enfant, vous sentez-vous plus stressée qu'à votre habitude? Pas du tout (1)–Un peu (2)–Moyennement (3)–Beaucoup (4)–Énormément (5). (In English: At present, due to your children's disorders, do you feel more stressed than usual? Not at all (1)–A bit (2)–Moderately (3)–A lot (4)–Enormously (5).)

was rephrased as follows in our questionnaire:

En raison du confinement, vous sentez-vous stressée? Beaucoup moins que d'habitude (1)– Un peu moins que d'habitude (2)–Comme d'habitude (3)–Un peu plus que d'habitude (4)– Beaucoup plus que d'habitude (5). (In English: Due to the lockdown, do you feel stressed?

Much less than usual (1)–A little less than usual (2)–As usual (3)–A little more than usual (4)–Much more than usual (5).)

For the purpose of this study, we analyzed responses relating to (1) the autistic individuals' nonfunctional socio-communicative behaviors and (2) their parents' QoL. Additionally, as mentioned above, we investigated the potential influence of four types of external factors by analyzing the effect of answers to questions related to: (1) home characteristics; (2) autistic individual characteristics; (3) resources (support and financial situation); and (4) educational strategies implemented at home.

## Participants

A total of 255 respondents fully completed the survey. Respondents were all parents (except for one grandparent) of an autistic individual and lived together with their autistic son or daughter regardless of his or her age. Among these, 43 were not Belgian residents at the time of the lockdown and were excluded from the final analyses to guarantee that all respondents underwent the same lockdown conditions. Two detailed questions were designed to control for the diagnosis of ASD: "Your autistic child was diagnosed by. . .? Autism Resource Center/ Psychiatrist/Neuropaediatrician/Pedopsychiatrist/Multi-disciplinary center/Diagnosis in process/No diagnosis/I do not know" and "What is your child's diagnosis? Autism Spectrum Disorder/Pervasive Developmental Disorder/Autism/Asperger Syndrome/I do not know". In three cases, strong doubts arose as to the validity of the diagnosis, which led to exclusion of the data because the respondent reported that the child did not have an ASD diagnosis or provided "I do not know" answers to those questions. A total of 209 respondents made up the final sample. Table 1 provides respondents' main descriptive statistics. It should be noted that age was split into four categories rather than used as a continuous variable. This choice is explained by the fact that those four categories reflect age categories used by Care and Support Services for ASD in Belgium.

**Table 1. Respondents' descriptive statistics.**

| Demographic data | N (209) | % |
|---|---|---|
| The autistic individual | | |
| *Gender* | | |
| Male:Female ratio | 156:53 | 74.64%:25.36% |
| *Age range* | | |
| 0-3y | 4 | 1.91% |
| 3-7y | 59 | 28.23% |
| 7-18y | 113 | 54.07% |
| +18y | 33 | 15.79% |
| *Comorbidities* | | |
| Genetic | 14 | 6.7% |
| Hyperactivity | 64 | 30.62% |
| Attention | 88 | 42.11% |
| Challenging behaviors | 112 | 53.59% |
| Sensory disorder | 67 | 32.06% |
| Epilepsy | 16 | 7.66% |
| Household information | | |
| *Outdoor space at the house* | | |
| None | 16 | 7.66% |

(*Continued*)

**Table 1.** (Continued)

| Demographic data | N (209) | % |
|---|---|---|
| Big | 153 | 73.21% |
| Small | 40 | 19.14% |
| *Self-reported COVID-19 positive case inside the household* | | |
| Negative:positive ratio | 193:16 | 93.34%:7.66% |
| Education and employment situation | | |
| *Respondent's education level* | | |
| Primary | 4 | 1.91% |
| Secondary | 87 | 41.63% |
| Tertiary | 118 | 56.46% |
| *Perception of financial situation before lockdown* | | |
| "Just enough" | 40 | 19.14% |
| "Fair" | 91 | 43.54% |
| "Comfortable" | 48 | 22.97% |
| "Insufficient" | 8 | 3.83% |
| "I do not want to answer" | 6 | 2.87% |
| "It's a daily challenging" | 16 | 7.66% |
| *Current working situation* | | |
| No change | 139 | 66.51% |
| Temporary unemployment or job loss | 24 | 11.48% |
| Teleworking | 46 | 22.01% |
| *Financial situation during lockdown* | | |
| Same | 133 | 63.63% |
| Less | 66 | 31.58% |
| More | 10 | 4.78% |
| *Alternative and continuity of professional support during the lockdown* | | |
| No | 91 | 43.54% |
| Yes | 100 | 47.85% |
| Not concerned | 18 | 8.61% |
| *Support by a new professional during lockdown* | | |
| No:Yes ratio | 193:16 | 92.34%:7.66% |
| Challenging behaviors during lockdown | | |
| *New hetero-aggressive behaviors (aggressions and violence towards others)* | | |
| No:Yes ratio | 155:54 | 74.16%:25.84% |
| *New auto-aggressive behaviors (self-harm)* | | |
| No:Yes ratio | 171:38 | 81.82%:18.18% |

## Data preparation

In order to interpret issues relevant to behavioral changes and quality of life, we computed two indexes by merging the Likert items answers to various questions from the survey addressing those two issues. Before creating these two variables, internal coherence of the pre-selected questions for each variable was assessed using the Cronbach's alpha (see Table 2).

The first index was called "nonfunctional socio-communicative behaviors"; such behaviors correspond here to various behaviors that aim to obtain attention and/or to express positive or negative feelings without using the adequate mode of verbal or non-verbal communication; this index was computed as the median answer per participant on 7 different Likert items (see Table 2). This measures the extent to which the lockdown brough about changes in

**Table 2. Components of the two variables and Cronbach's alphas.**

| Variable name | Components of the variable (answers on a Likert item from 1 to 5) | Cronbach's alpha |
|---|---|---|
| Autistic individuals' nonfunctional socio-communicative behaviors | Since lockdown, does your child exhibit hetero-aggressive behaviors (e.g., to bite, to hit, to break, to throw, etc.)? | 0.77 |
| | Since lockdown, does your child exhibit auto-aggressive behaviors (e.g., to bite oneself, to hit oneself, etc.)? | |
| | Since lockdown, does your child exhibit restricted and repetitive behaviors when he is anxious? | |
| | Since lockdown, does your child seek for attention and interaction (in any way)? | |
| | Since lockdown, does your child seek for attention and interaction without you understanding why? | |
| | Since lockdown, does your child seek for hugs and physical interaction? | |
| | Since lockdown, does your child exhibit temper outbursts at home (in comparison to a usual day prior to lockdown)? | |
| Parents' quality of life | Since lockdown, you worry... | 0.85 |
| | Since lockdown, you feel upset... | |
| | Since lockdown, you feel stressed... | |
| | Since lockdown, do you easily lose your temper? | |
| | Is your mood affected by lockdown? | |
| | Since lockdown, do you feel disarmed? | |
| | Since lockdown, do you feel overwhelmed by the daily grind? | |
| | Since lockdown do you restrict yourself from going out? | |
| | Since lockdown, is your family life conflictual? | |
| | Since lockdown, has your couple life been affected? | |
| | Since lockdown, do you feel in harmony with your partner for the daily life intendancy? | |
| | Since lockdown, does your child with ASD wake you up at night? | |

nonfunctional socio-communicative behaviors, with higher scores corresponding to more frequent occurrences than usual and lower scores to less frequent occurrences than usual.

The second variable was "Parents' QoL" and was computed as the median answer per participant on 12 different Likert items (see Table 2) addressing issues of parents' QoL. The index of parents' QoL also measures the presence of stress and QoL-related feelings due to the restrictions, so that higher scores on the Likert scale indicate a drop in QoL.

## Analytic plan

All statistical analyses were conducted in R [39]. The dependent variables in all reported analyses were either the index of autistic individuals' nonfunctional socio-communicative behaviors or the index of parents' quality of life. The impact (positive or negative) of lockdown on the two indexes of autistic individuals' nonfunctional socio-communicative behaviors and parents' QoL was analyzed using One-Sample Wilcoxon Signed Rank Tests using the wilcox.test function and Spearman correlations using the cor.test function. Subsequently, the potential impact of each of four domains (Home characteristics, Autistic individuals' characteristics, Resources, Home educational strategies) on the autistic individuals' nonfunctional socio-communicative behaviors and their parents' QoL during lockdown was investigated with linear regressions, implemented with the lm function. Each domain was composed of a set of predictors (fixed effects) chosen a priori and detailed in the corresponding subsections of the Results section. The independent effect of the predictors was further analyzed only when the domain as a whole (the entire set of predictors) had an effect on the dependent variable, i.e. when the simple linear regression model reached significance.

## Results

### Impact of lockdown on the autistic individuals' behaviors and their parent's quality of life

Fig 1, which displays stacked bar charts for the two variables clearly shows that lockdown restrictions had a negative impact for many autistic individuals and their parents. For a vast majority of autistic individuals and parents, when government restrictions changed their behaviors and QoL respectively, it was in a negative way. Sixty-five percent of respondents reported a poorer QoL since lockdown and 47% of them reported that their child exhibited more nonfunctional socio-communicative behaviors.

One-Sample Wilcoxon Signed Rank Tests confirmed that responses on the Likert scales were significantly higher than 3 ("As usual") for autistic individuals' nonfunctional socio-communicative behaviors ($z = -8.37$, $p < .001$) and for parents' QoL ($z = -8.27$, $p < .001$). There was a significant positive association between autistic individuals' nonfunctional socio-communicative behaviors and parents' QoL (Spearman $r = .44$, $p < .001$), which indicated that an increase of such behaviors correlated with a lower QoL than usual.

### What had an influence on the impact of lockdown on the children's socio-communicative behaviors and their parent's quality of life?

Next, we analyzed the impact that each domain had on the autistic individuals' nonfunctional socio-communicative behaviors and parents' QoL by implementing linear regression on the two indexes. S1 and S2 Tables in S1 File summarize the results of the simple linear regression models.

**Domain 1: Home characteristics.** Number of cohabitants, Number of minor cohabitants, Bedrooms missing (Yes, No), Type of housing (Appartement, Semi-detached house, Detached house), Outside space at home (Absent, Big, Small) were used as fixed effects for the autistic individuals' behaviors. All these same variables, with the addition of Family composition (Parental couple, Single-parent family, Stepfamily) were used as fixed effects to regress the parents' QoL. There was no overall effect of the home characteristics on the autistic

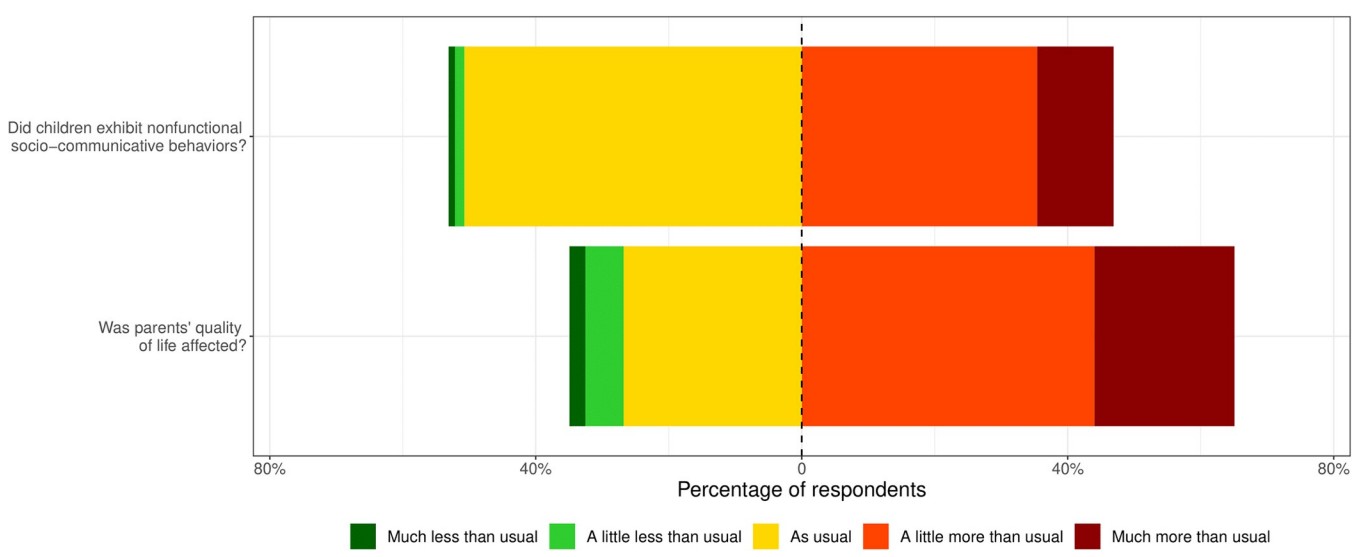

**Fig 1. Distribution of the median responses on the nonfunctional socio-communicative behaviors and quality of life Likert scales.**

individuals' socio-communicative behaviors ($F_{(7, 201)}$ = 1.7, $p$ = .11, $R^2$ = .06) and on parents' QoL ($F_{(9, 199)}$ = 1.05, $p$ = .4, $R^2$ = .04). In sum, there was no association between the characteristics of respondents' households and the impact of lockdown restrictions on autistic individual's behaviors and on the QoL of their parents.

**Domain 2: Autistic individuals' characteristics.**    Age (0 to 3, 3 to 7, 7 to 18, More than 18), Gender (Female, Male), Number of comorbidities and Education (Special education school, Regular education) were used as fixed effects for the autistic individuals' behaviors and the parents' QoL. There was a significant overall effect of the autistic individuals' characteristics on autistic individuals' nonfunctional socio-communicative behaviors ($F_{(6, 202)}$ = 2.74, $p$ = .01, $R^2$ = .07). Once each predictor was individually considered, however, only Number of comorbidities was a significant predictor in this model ($\beta$ = .11, SE = .04, $p$ = .008). Autistic individuals with more comorbidities exhibited more nonfunctional socio-communicative behaviors during the lockdown period. By contrast, there was no overall effect of the autistic individuals characteristics on parents' QoL ($F_{(6, 202)}$ = 1.35, $p$ = .24, $R^2$ = .04).

**Domain 3: Resources.**    Parents' access to respite care (on a 5-point Likert item), Financial worries during lockdown (on a 5-point Likert item), Partner's support (on a 5-point Likert item), Professional support (on a 5-point Likert item), Parents' group support (on a 5-point Likert item) were used as fixed effects for the autistic individuals' behaviors. One additional fixed effect was used for the parents' QoL: Contacts with friends during lockdown (on a 5-point Likert item).

Results of the simple linear regression indicated a significant overall effect of resources variables on the autistic individuals' nonfunctional socio-communicative behaviors ($F_{(5, 203)}$ = 4.68, $p$ < .001, $R^2$ = .1). Further analysis revealed a significant effect of Parents' access to respite care ($\beta$ = -.19, SE = .05, $p$ < .001). Parents with less access to respite care were more likely to report an increased amount of their child's nonfunctional socio-communicative behaviors during the lockdown period.

There also was an overall effect of resource variables on parents' QoL ($F_{(6, 202)}$ = 7.63, $p$ < .001, $R^2$ = .18). Further analysis revealed a significant effect of Parents' access to respite care ($\beta$ = -.31, SE = .06, $p$ < .001). Parents with less access to respite care were more likely to report a lower QoL during the lockdown period.

**Domain 4: Home educational strategies.**    Homework from school (Yes, No), Implementation of homeschooling (Yes, No), Schedule at home (New schedule, Same schedule, No schedule), Implementation of rules at home (on a 5-point Likert item), Implementation of ASD strategies (on a 5-point Likert item) were used as fixed effects to regress the two indexes.

There was a significant overall effect of home educational strategies on autistic individuals' nonfunctional socio-communicative behaviors ($F_{(6, 202)}$ = 2.57, $p$ = .02, $R^2$ = .07). Further analyses revealed a significant effect of Implementation of rules at home ($\beta$ = -.12, SE = .14, $p$ = .03). Autistic individuals who lived in a home with less rules were reported to have exhibited more nonfunctional socio-communicative behaviors during the lockdown period. Finally, there was no effect of home educational strategies on parents' QoL ($F_{(6, 202)}$ = .59, $p$ = .74, $R^2$ = .02).

## Discussion

The present study investigated how the COVID-19 lockdown altered the frequency of nonfunctional socio-communicative behaviors in autistic individuals and their parents' QoL. Targeting French-speaking Belgian residents, we conducted a parental questionnaire study to collect such highly contextualized data. This study thus adds to the literature on the impact of COVID-19 on the autism community, first, by focusing on the French-speaking Belgian

situation, and second, by exploring how the impact of lockdown could be influenced by the characteristics of the autistic individuals and those of their household, by the availability of financial and support resources, and by the implementation of educational strategies.

## Impact of lockdown on the autistic individuals' behaviors and their parent's quality of life

Consistent with previous studies conducted in other countries [10, 13, 40], our results indicate that lockdown restrictions were associated with negative changes in autistic individuals' socio-communicative behaviors. We speculated in the introduction that one could reasonably hypothesize that the lockdown restrictions may have brought about positive changes for some autistic individuals. Our results, however, unequivocally show that this was not the case in our sample. Parents reported that nonfunctional socio-communicative behaviors were significantly more frequent than before lockdown. These results are remarkably similar to those by Colizzi and collaborators [10], in whose sample 41.5% of parents of autistic individuals reported higher frequency of behavior problems during the Italian lockdown (as compared to 47% of our sample).

Therefore, the autistic population was no exception to the generally negative impact of lockdown restrictions on the population [41]. The lockdown is likely to have drastically intensified the probability of losing routines and reference points, and of changing habits and daily rhythms. Combined with the low tolerance for uncertainty and insistence on sameness, frequently attested in ASD, lockdown restrictions probably generated higher than usual levels of stress and anxiety. There is, indeed, evidence that stress and anxiety can be a source of repetitive challenging and nonfunctional communicative behaviors in ASD [42, 43].

Our results indicate that parental QoL, which is already fragile in families of autistic children (4–7), worsened following lockdown restrictions, as 65% of the respondents reported a negative change in their QoL during the lockdown period, confirming previous findings that the lockdown had a negative impact on parents of autistic children [44]. Outside the context of the pandemic, parental QoL has been showed to be negatively impacted by the autistic child's characteristics, such as the presence of challenging behaviors and other comorbidities [45]. As shown above (see Fig 1), 47% of our respondents reported that their autistic child displayed more nonfunctional socio-communicative behaviors since the lockdown onset, which is very likely to have had an impact on parental QoL. Previous studies also report several protective factors for parental QoL such as the autistic child's level of socio-communicative skills, high family income, social support or the presence of professional interventions from which the child benefits [38, 46, 47]. Most of these protective factors were probably reduced by the lockdown restrictions. A correlation between autistic individuals' nonfunctional socio-communicative behaviors and their parents' QoL is well documented in the literature [6, 38] and was replicated during the pandemic context in the present study. In line with Alhuzimi's [15] and Nonweiler and collaborators' [40] findings for Saudi Arabian and British populations, our results show that the frequency of autistic individuals' nonfunctional socio-communicative behaviors was positively associated to the degradation of parents' QoL during lockdown.

We also investigated which factors may correlate with the impact of lockdown on the children's behaviors and their parent's quality of life.

## Household characteristics

None of the Household variables we tested had a significant impact on the increase of the autistic individuals' nonfunctional socio-communicative behaviors or on parental QoL. This may be surprising, as there is a well-documented correlation between housing type and the

mental and emotional well-being of inhabitants [48]. The availability of an outside space at home can have a protective effect for children's well-being and prevent behavioral issues [49], contribute to better emotional and behavioral outcomes [50] or improve autistic individuals' well-being [51]. Note, however, that our results mirror those by Westrupp and collaborators [44], whose research during the Australian lockdown found no impact of the availability of outdoor space on autistic children's or their parents' mental health. Additionally, Westrupp et al. [44] found, just like us, that neither being a single parent nor the number of children in the household had an effect on parents during the lockdown.

## Autistic individuals' characteristics

The only autistic individuals' characteristic that significantly predicted an increase in nonfunctional socio-communicative behaviors was the reported number of comorbidities. Regardless of age or gender, the more comorbidities the autistic individual had, the higher the increase in nonfunctional socio-communicative behaviors displayed during the lockdown.

Our results are consistent with previous studies on the presence of challenging behaviors, which did not find a clear association between challenging behaviors, socio-communicative behaviors and age or gender [52–54]. Furthermore, Colizzi et al. [10] and Alhuzimi [15] also report no significant effect of age on the frequency of the child's behavioral problems during the lockdown. Our result that the number of comorbidities is a significant predictor of an increase in nonfunctional socio-communicative behaviors during lockdown is in line with recent findings from studies on the COVID-19 pandemic situation [30, 36], and more generally, with studies that report overall higher rates of nonfunctional socio-communicative or challenging behaviors for autistic individuals with a higher number of comorbid problems [55].

None of the autistic individuals' characteristics had an effect on parents' QoL. In line with these results, Derguy et al. [56] found no relation between child characteristics (sex or gender) and parents' QoL, although they did report that a higher number of comorbidities was associated with a low parental QoL. At least during the lockdown, autistic individuals' characteristics did not seem to impact the significative drop in parental QoL. However, this is in contrast with Manning and collaborators [12] who found a negative correlation between the autistic individual's age and stress for caregivers.

## Resources

Availability of parental respite care had a positive effect on autistic individual's socio-communicative behaviors and on parents' QoL during lockdown. Although the benefits of respite care for parents of children with developmental disabilities is well attested [57], and confirmed by the systematic review by Cooke and collaborators [58] already before the COVID-19 pandemic, the need for such resources are still largely unmet, at least for French-speaking Belgian residents.

Surprisingly, the relationship between availability of social and professional support and parents' QoL did not emerge in our study, in spite of having been reported in several studies prior to the COVID-19 pandemic [45]. During lockdown restrictions parents probably need more respite care than professional or family support, the access to which was in any case highly restricted or even impossible.

## Home educational strategies

Finally, only the implementation of rules at home had a significant effect on autistic individuals' nonfunctional socio-communicative behaviors. By contrast, we found no effect of the

implementation of ASD strategies or the use of schedule. During lockdown, autistic individuals who lived in a home with less rules than usual were reported to have displayed more non-functional socio-communicative behaviors. Interestingly, these results suggest that rather than the type or the specificity of strategies, it is the continuity and the consistency of the rules at home over time that may have a significant impact on autistic individuals' socio-communicative behaviors.

## Limitations and future directions

This study focuses on one particular time period of the COVID-19 pandemic. Our observations only apply to that specific period and future studies could investigate whether these results can be replicated throughout the whole pandemic and whether families fully recover their pre-pandemic balance once the pandemic is over. That said, the specificity of our population does not make the results any less important or interesting. Moreover, our sample includes many participants on the spectrum with different profiles, reflecting the heterogeneity of the spectrum. The study therefore contributes and has added value when taken together in the larger scope of the few studies on the COVID-19 impact on autistic individuals. By going in the same direction as other studies (i.e., negative impact), it further validates previously reported results. Replicability over different regions and populations is essential for the robustness of reported results. The fact that a negative impact is found across regions and populations gives more strength to the result.

As mentioned in the Methods section of this paper, the questionnaire was widely disseminated in the French-speaking Belgian autistic community during the first strict lockdown. However, we cannot rule out the fact that some of the target population may not have been able to take the time to answer our online questionnaire.

Another limitation is that our questionnaire is new and has not been validated before this study. However, the internal consistency is high between the items reported in this paper, and the part of the questionnaire targeting parental QoL is based on a previously validated tool [38]. The use of a questionnaire also implies that we rely solely on parental reports. Information reported by the parents can obviously be influenced by their state of mind and recollection of events. Additionally, key information such as diagnostic status could not be confirmed by a direct assessment and relied only on parental reports. However, parental reporting is probably one of the best tools to assess and report changes in their child's daily behaviors and their own QoL.

Since this first lockdown, autistic people and their families have had to get used to living with fluctuating and recurrent restrictions on social life. There is a need to pursue long-term follow-up of autistic people during and after the pandemic, and more generally, when their routines are disrupted (e.g., long summer breaks). In particular, there are outstanding questions on services, training programs and alternative ways (e.g., digital adaptations) to support populations with special needs when their social contacts are restricted or drastically modified when, for example, schools close during breaks or ends for young autistic adults.

## Conclusion

It is a truism that the COVID-19 outbreak, and the ensuing governmental lockdown restrictions created an unpreceded situation. The few studies available that were implemented during the lockdown report a general negative effect of the COVID-19 outbreak and lockdown on autistic individuals and their parents. The present study adds to this still limited literature by focusing on the effects of lockdown on autistic individuals' socio-communicative behaviors and their parents' quality of life in French-speaking Belgian residents. Lockdown restrictions

were associated with an increase in the autistic individuals' nonfunctional socio-communicative behaviors and a decrease in their parents' quality of life. The negative predictors of the impact of COVID-19 highlighted in this study are the number of comorbidities the autistic individual has, the non-implementation of new rules at home during the lockdown, and the lack of access to respite care for parents.

## Supporting information

**S1 File.**
(DOCX)

## Acknowledgments

We express our gratitude to all the families that took part in this study sharing their experience during this hard time. We also thank Eric Willaye, the SUSA Foundation director, for insights and support throughout the research, as well as Aurore Delhommeau, Aurélie Hanquet and Lucie Ducloux for the help in designing the questionnaire. Finally, we would like to thank Eleanor Miller for her corrections on our manuscript.

## Author Contributions

**Conceptualization:** Marielle Weyland, Pierre Defresne.

**Data curation:** Marielle Weyland, Pauline Maes.

**Formal analysis:** Marielle Weyland, Pauline Maes.

**Funding acquisition:** Mikhail Kissine.

**Investigation:** Marielle Weyland.

**Methodology:** Marielle Weyland, Pauline Maes, Pierre Defresne.

**Project administration:** Marielle Weyland.

**Software:** Marielle Weyland.

**Supervision:** Marielle Weyland, Mikhail Kissine, Pierre Defresne.

**Visualization:** Pauline Maes.

**Writing – original draft:** Marielle Weyland, Pauline Maes, Pierre Defresne.

**Writing – review & editing:** Marielle Weyland, Pauline Maes, Mikhail Kissine, Pierre Defresne.

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
