## [Decision Letter · Decision Letter 0]

25 Jul 2022

PONE-D-22-09618Impact of Belgian COVID-19 Lockdown Restrictions on Autistic Individuals’ Socio-communicative Behaviors and their Parents' Quality of LifePLOS ONE

Dear Dr. Weyland,

Thank you for submitting your manuscript to PLOS ONE. After careful consideration, we feel that it has merit but does not fully meet PLOS ONE’s publication criteria as it currently stands. Therefore, we invite you to submit a revised version of the manuscript that addresses the points raised during the review process.

Please address the minor issues raised by the reviewers. ==============================

We look forward to receiving your revised manuscript.

Kind regards,

Daswin De Silva

Academic Editor

PLOS ONE

“This work is supported by various sources of funding. MW and PD are supported by research grants from the Support Fund Marguerite-Marie Delacroix. PM is supported by a doctoral grant from the ROGER DE SPOELBERCH Foundation. MK is a 2019-2021 Francqui Foundation Research Professor”

“We express our gratitude to all the families that took part in this study sharing their experience during this hard time. We also thank Eric Willaye, the SUSA Foundation director, for insights and support throughout the research, as well as Aurore Delhommeau, Aurélie Hanquet and Lucie Ducloux for the help in designing the questionnaire.”

“This work is supported by various sources of funding. MW and PD are supported by research grants from the Support Fund Marguerite-Marie Delacroix. PM is supported by a doctoral grant from the ROGER DE SPOELBERCH Foundation. MK is a 2019-2021 Francqui Foundation Research Professor”

Reviewers' comments:

Reviewer's Responses to Questions

**Comments to the Author**

1. Is the manuscript technically sound, and do the data support the conclusions?

Reviewer #1: Yes

Reviewer #2: Yes

2. Has the statistical analysis been performed appropriately and rigorously? 

Reviewer #1: I Don't Know

Reviewer #2: Yes

3. Have the authors made all data underlying the findings in their manuscript fully available?

Reviewer #1: Yes

Reviewer #2: Yes

4. Is the manuscript presented in an intelligible fashion and written in standard English?

Reviewer #1: Yes

Reviewer #2: No

5. Review Comments to the Author

Reviewer #1: The introduction has a nice flow, provides useful context and sets the scene well.

Line 78 and 80: "or in Michigan" and "or in self reports". I believe the "or" in both instances are typos and should be replaced with "and"

Line 102: the word "other" appears twice in this sentence. Please re-word the sentence to address this.

Line 142: Since data collection has been completed, please change sentence to past tense.

Line 156: Please check for typo/grammar errors. e.g. "child" in this sentence should be replaced by "child's"

Line 164: "in 15 subdomains" - should read "into 15 subdomains"

Line 170 - 172: This sections, especially the part on 15 subdomains feels repetitive (ref Line 164). Consider re-wording.

Line 173-174: Not sure if using a new paragraph was intentional, but feels unnecessary.

Line 221: "makes up the final sample.." . Please check the tense and ensure it is consistent throughout the paper and presented in past tense as the work has been completed.

Table 1 is clearly presented and provides a good description of the sample, and variables explored. Nice work!

Line 326: Please use consistent terminology for this domain. I feel it is best to refer to it as labelled in a preceding section of the paper, "Home education strategies".

Line 369: I question if "paid a tribute" is the most appropriate wording to use in this context?

Line 469" replace "event" with "events"

Line 470: replace "rely" with "relied"

Line 472- 479: I question the relevance of this section as it is clear, a comparison with the counterfactual is not going to yield an answer to the research question. Consider if inclusion is therefore required.

Line 482-484: Can the authors please consider including descriptors of the relevance of the findings to additional contexts, apart from when social contacts are restricted? Covid 19 disrupted routines, accessibility to support for caregiving etc. This concluding statement on the relevance of the findings should point to supporting families experiencing these scenarios as well.

The topic is interesting and relevant. The findings make an important contribution to the limited evidence base. The paper has been written exceptionally well, and was a pleasure to read.

Reviewer #2: This paper aimed to assessed the impact social restrictions may have on autistic individuals’ behavior and their parents’ or caregivers’ quality of life. It is relevant to the aims and scope of the journal. Just a few minor revisions along the following points before it can be considered:

1. Kindly describe how were the psychometric properties of the instrument ensured provided that some modifications were made to it.

2. There are few but significant grammatical errors; thorough language-editing should be made.

Thanks!

6. PLOS authors have the option to publish the peer review history of their article (what does this mean?). If published, this will include your full peer review and any attached files.

Reviewer #1: No

Reviewer #2: No

---

## [Author Response · Author response to Decision Letter 0]

27 Jul 2022

Our Reponse to reviewers has been uploaded as .doc file with the other documents of our submission.

---

## [Editor Report · Decision Letter 1]

18 Aug 2022

Impact of Belgian COVID-19 lockdown restrictions on autistic individuals’ socio-communicative behaviors and their parents' quality of life

PONE-D-22-09618R1

Dear Dr. Weyland,

We’re pleased to inform you that your manuscript has been judged scientifically suitable for publication and will be formally accepted for publication once it meets all outstanding technical requirements.

Kind regards,

Daswin De Silva

Academic Editor

PLOS ONE
---

## [Editor Report · Acceptance letter]

19 Aug 2022

PONE-D-22-09618R1 

Impact of Belgian COVID-19 lockdown restrictions on autistic individuals’ socio-communicative behaviors and their parents' quality of life 

Dear Dr. Weyland:

I'm pleased to inform you that your manuscript has been deemed suitable for publication in PLOS ONE. Congratulations! Your manuscript is now with our production department. 

Kind regards, 

on behalf of

Dr. Daswin De Silva 

Academic Editor

PLOS ONE